# Enhancing the Performance of SQL Injection Attack Detection through Probabilistic Neural Networks

Fawaz Khaled Alarfaj [1,*] and Nayeem Ahmad Khan [2]

1 Department of Management Information Systems, School of Business, King Faisal University, Al-Ahsa 31982, Saudi Arabia
2 Faculty of Computer Science & Information Technology, AlBaha Al-Baha University, Al-Baha 65779, Saudi Arabia
* Correspondence: falarfaj@kfu.edu.sa

**Abstract:** SQL injection attack is considered one of the most dangerous vulnerabilities exploited to leak sensitive information, gain unauthorized access, and cause financial loss to individuals and organizations. Conventional defense approaches use static and heuristic methods to detect previously known SQL injection attacks. Existing research uses machine learning techniques that have the capability of detecting previously unknown and novel attack types. Taking advantage of deep learning to improve detection accuracy, we propose using a probabilistic neural network (PNN) to detect SQL injection attacks. To achieve the best value in selecting a smoothing parament, we employed the BAT algorithm, a metaheuristic algorithm for optimization. In this study, a dataset consisting of 6000 SQL injections and 3500 normal queries was used. Features were extracted based on tokenizing and a regular expression and were selected using Chi-Square testing. The features used in this study were collected from the network traffic and SQL queries. The experiment results show that our proposed PNN achieved an accuracy of 99.19% with a precision of 0.995%, a recall of 0.981%, and an F-Measure of 0.928% when employing a 10-fold cross-validation compared to other classifiers in different scenarios.

**Keywords:** cybersecurity; SQL injection attacks; deep learning; probabilistic neural network; BAT optimization algorithm

## 1. Introduction

The IT infrastructure is experiencing swift growth, and our dependence on technology is increasing at an unprecedented pace, fueled by exponential expansion. This growth has revolutionized the way we live and work, bringing with it both opportunities and challenges. Nearly all of the applications that we use in our daily life are web-based. These applications are made available online without borders to make human lives better. Since these applications can be accessed from any part of the world, this brings a security challenge in the form of uncontrolled and unauthorized access [1]. While using these online web applications or websites, data are generated from the user stored in these web applications or the website's backend database. These databases are managed by DBMS, which work by running the statements or queries written in SQL. The front-end application and backend databases are vulnerable to many types of attacks as they are being accessed through the internet. Among the reported vulnerabilities by OWASP, SQL injection attacks are included in the top 10 vulnerabilities [2]. According to recent trends, the attack rate has increased by over 300% over the last ten years, with attackers using sophisticated methods, such as obfuscated code and encryption to avoid detection [3]. An SQL injection attack is performed by injecting or inserting an SQL query through the data input field from the client side into the web application or website [4]. The user input query can be malicious, and if the query is executed successfully it may lead to modifying the database, spoofing identity, leaking sensitive information, and causing repudiation [5].

SQL injection attacks are one of the most dangerous flaws that attackers use to obtain private data without permission, steal money, and hurt reputations [6]. Information disclosure, data manipulation, account takeover, code execution, and denial of service are some of the most hazardous vulnerabilities exploited as a result of SQL injection attacks. Attackers may access and reveal sensitive data, including usernames, passwords, credit card numbers, and personal information via SQL injection attacks. This data can then be exploited for identity theft, fraud, and other nefarious purposes. Moreover, they have the ability to alter, remove, or even seize control of all of the database's data, which might result in the loss of crucial information as well as financial and reputational harm. Moreover, attackers have the ability to run malicious code on the targeted machine, obtain remote access to it, and take control of it. This allows for the theft of confidential data, data destruction, and other nefarious actions. Moreover, SQL injection attacks may be used to conduct denial-of-service attacks by bombarding the targeted system with many SQL queries, resulting in the system crashing or going down and inflicting harm to the system's reputation and finances [7,8]. By adopting security best practices, performing routine vulnerability assessments, and using sophisticated detection and prevention methods, such as machine learning-based approaches, companies may take proactive steps to stop and identify SQL injection attacks.

Web applications need to learn and sanitize the user input from external sources for any possible malicious SQL injection. Obtaining unauthorized access may be due to software developers' poor programming practices and the cybercriminals who intentionally design the attack behind the scenes [9]. Cybercriminals regularly work on practices to get hold of sensitive data to achieve their objectives. Four types of SQL injection attacks are Error based SQL injection, UNION based SQL injection, Blind SQL injection, and Out of band SQL injection [10]. The Out of band SQL type is termed one of the common attacks, occurring when an invalid common is entered as an input, which triggers the response from the database server. The database server may respond with an error message containing the details about the data structure, version of DBMS, type and version of operating stem running and, in some cases, display the complete query results. UNION-based SQL injection attacks use the UNION operator of SQL, allowing multiple statements to be combined into at least one among the combined query performing the malicious activities by extending the output generated by the original query. In a Blind SQL injection attack, a query is inserted, requiring the database to respond to the true or false question [11]. Attackers analyze the response to a question and can obtain sensitive information. Out of band SQL injection is a very uncommon attack type, for the most part; it relies on features allowed on the database server to be utilized by the web application. Fundamentally, this attack is performed when an attacker is inept at utilizing the same attack channel to obtain the results.

In recent times, deep learning, a subset of machine learning, has played an essential role in the classification of natural language processing and image pattern recognition [12]. Several studies have been conducted on how to benefit from machine learning in cybersecurity [13–15]. A neural network, a part of deep learning that imitates the human brain by making a network of neurons that are combined and modelled as a neural network, has been applied in detecting malicious code attacks [16]. The significant advantages provided by the neural network have led to the elimination of machine learning problems by increasing detection accuracy, and many studies have reported the same.

To improve the accuracy of SQL injection attacks, researchers have developed various techniques, including signature-based detection and anomaly-based detection [17]. However, these techniques often have limitations in their ability to accurately detect and prevent SQL injection attacks. The main contribution of this work is the development of a PNN-based model for detecting SQL injection attacks. The model is trained using a large dataset of SQL queries, both legitimate and malicious, and is able to accurately distinguish between the two types of queries. This allows the model to identify and block SQL injection attacks with a high degree of accuracy, while minimizing false positives. Our experiments

uncover how PNN performed better when smoothing parameters were selected using the BAT optimization algorithm. Therefore, it establishes that the proposed approach can effectively detect SQL injection attacks.

The rest of the paper is organized as follows: Section 2 provides the details about related work. Section 3 details the overview of the proposed approach for detecting SQL injection using a Probabilistic Neural network. In Section 4, the experimental results and implementation details are provided. Section 5 concludes the work.

## 2. Related Work

The majority of conventional SQL injection attack detection methods depend on anomaly- and signature-based detection techniques. The use of pre-established patterns, rules, or signatures to identify known SQL injection threats is known as signature-based detection [18,19]. This strategy often entails constructing a database of known SQL injection attack patterns and evaluating the incoming traffic against this database to detect possible attacks. Nevertheless, signature-based detection has drawbacks since it cannot identify attacks that were previously undetected and is readily thwarted by attackers who alter the attack pattern.

Conventional SQL injection attack detection approaches also include the use of two input validation methods, such as whitelist and backlist [20]. Whitelisting works by filtering the content of the input. If any character or set of characters is present in the SQL query, such a query will be blocked for further processing. Although whitelisting works inversely to blacklisting, if a character or set of characters is absent, such a query will be blocked from further processing. The major challenge with such validation methods is that they cannot detect new SQL injections, and lists need frequent updating. The amount of time between the detection of a new SQL injection query and updating the list may be long, and cybercriminals may benefit from the time gap and may perform the attack.

Several studies have applied machine learning for the detection of malware attacks. Among the most popular include SQL injection, cross-site request forgery, cross-site scripting, distributed denial of service, broken authentication, phishing, etc. A study by [21] proposed an approach for SQL injection that employs a Naive Bayes classifier built on role-based access control. The experimental results show an accuracy of 93.3% with a precision of 1.0% and recall of 0.89%. The study is limited only to a specific type of SQL injection attack. A study by [22] proposed an approach for the prevention of SQL injection by using four different classifiers, such as Support Vector Machines (SVM), Artificial Neural Networks (ANN), Boosted Decision Tree and Decision Tree. In this study, the authors have used a database of 1100 vulnerable samples of SQL injection. The comparison of results among the four classifiers shows that the Decision Tree performed better. The downside of this approach is its huge processing time. A study by [23] proposed REGEX, a regular expressions filter approach for SQL injection attack detection. In this study, a dataset of 20,474 queries was used. The downside of using the regular expressions filter is that it cannot detect novel SQL injection attacks. A few studies have been conducted using artificial neural networks, including a study by [24] that proposed an approach based on deep learning and random projection to detect malicious JavaScript. For feature extraction, denoising auto-encoders were used. This study is limited to the detection of only malicious JavaScript attacks. A study by [25] proposed and implemented a Fuzzy neural network to build an expert system for SQL injection attack detection. Expert systems are complex in their development and time-consuming. A study by [26] proposed an approach called CODDLE to detect malicious injection attacks based on deep learning. A Convolutional Deep Neural Network was used, and the new encodes of SQL/XSS symbols were at the pre-processing stage. Experimental results show that an accuracy of 95% was achieved, along with a precision of 99% and a recall value of 92%.

## 3. Detection of SQL Injection Using Probabilistic Neural Network

The presented method of detecting SQL injection attacks is based on probabilistic neural networks to achieve the highest detection accuracy. The methods employ a self-learning approach that is adept in the detection of unknown SQL injection attacks. The self-learning approach, coupled with deep learning, enables the selection of complex features from an SQL query to distinguish between malicious and benign SQL queries. The steps involved in this proposed approach are presented in Figure 1.

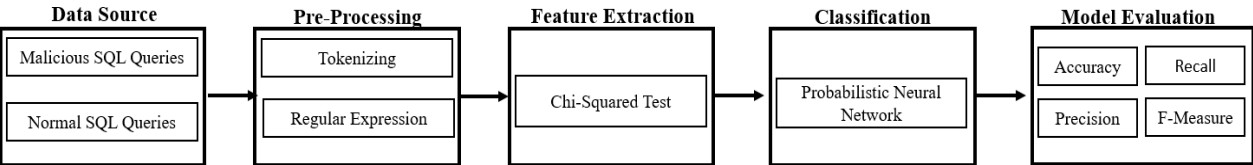

**Figure 1.** Architecture of the Proposed Approach.

### 3.1. Data Collection

SQL injection datasets are not publicly accessible, and some organizations are not willing to share the dataset. In this approach, we collected data from a python library known as Lib-injection, which contains all forms of SQL injections [27]. We collected 6000 SQL injection queries and 3500 normal SQL queries.

### 3.2. Pre-Processing

The next stage after the data collection is pre-processing. Data pre-processing is the crucial stage in developing a machine learning-based model. Data pre-processing enables the cleaning and preparing of the data from the raw data and making it fit for the model. During pre-processing the data are standardized, noise and repeated data are removed, and missed values are identified. In this study, we performed tokenizing on the SQL queries where the query is broken down into strings, such as keywords, symbols, phrases, and punctuation marks, also called tokens. In this approach to detect the SQL injection, each character is kept, and the token is produced through regular expressions. We utilized the RegEx module of Phyton for regular expressions [28].

### 3.3. Feature Extraction

Feature extraction, a dimensionality reduction, helps identify important features in the given feature space. It is a known rule of machine learning that not all of the extracted features will contribute to achieving the accuracy required to build an efficient model. In this study, we use the Chi-Square test for the feature section. A Chi-Square test is a statistical test used to test the independence of two events. Based on the two variables, the observed O and expected E Count are computed. Chi-Square calculates the deviations of E from O. Equation (1) represents the formula for the Chi-Square. The advantages of using Chi-Square are its robustness regarding the data distribution, it is computational and it extracts comprehensive test information. The features used in this study were collected from the network traffic, and the SQL queries are given in Table 1. The extracted set of features has never been used in the detection of SQL injection attacks. c denotes the degree of freedom.

$$x_c^2 \sum \frac{(o_i - E_i)^2}{E_i} \tag{1}$$

**Table 1.** List of Selected Features.

| Feature Name | Feature Description |
| --- | --- |
| Token_Count | Counts the presence of specific tokens in the whole dataset |
| Token_Type | Category to which the token belongs, such as plain text or SQL injection attack |
| Token_Value | The actual parameter value of the token |
| Type_Protocol | Transmission Control Protocol (TCP) Internet Protocol (IP) Internet Control Message Protocol (ICMP) |
| Length_TCP | Data Offset; The size of the TCP header in 32-bit words |
| http_content_length | Length of Entity Body |
| Type_Port_Req | Port Type |
| Source_Data_length | Number of Characters in a String |

### *3.4. Probabilistic Neural Network*

A neural network consists of interconnected layers of artificial neurons. The neurons in the input layer are connected to the neurons in the hidden layer, which are then connected to the output neuron. The weight of each connection represents how much impact that particular connection has on a neuron's activity. One of the most popular neural networks is called a probabilistic neural network (PNN) [29]. PNNs are based on Bayesian inference and are used for classification and regression tasks, such as predicting prices or ranking search results. Every time the PNN model creates a piece of text, it becomes better at predicting which words are likely to come next. This process is similar to the way humans learn language and grammar from the environment. The PNN predicts the value of a variable by combining information from other variables in an inferred probabilistic way [30]. The kernel performs the basic operation in PNN. It is used to compute the probability of a particular outcome given an input. A kernel can be seen as a function that takes an input and maps it to another dimension (feature map). The kernel is then multiplied by all the dot products with each feature map. The advantage of using PNN is that it is significantly faster in training, produces more accurate results than multilayer perceptron networks, and is relatively insensitive to outliers. In view of the advantages of PNN, we extend it to the detection of SQL injection attacks.

PNNs have four types of layers, an input layer, a pattern layer, a summation layer, and an output layer; inputs flow through a series of "hidden" layers before reaching an output level, as shown in Figure 2 [31]. The function of the input layer is to obtain the input and distribute it to the neurons. Once the input is delivered from the input layer to the pattern layer in the form of a pattern $x_{ij}$, which is the neuron vector, the output is computed by the pattern layer using Equation (2).

$$\varnothing_{ij}(x) = \frac{1}{(2\pi)^{\frac{d}{2}} \sigma^d} \exp[\frac{(x - x_{ij})^T (x - x_{ij})}{2\sigma^2}] \qquad (2)$$

where *d* represents the dimension related to the pattern vector *x*, and *σ* represents the parameter for smoothing. The summation layer performs the estimation of the maximum likelihood of pattern *x* being categorized in $C_i$ classes. The outputs produced by neurons are combined and averaged according to which particular class they belong. The summation layer is based on Equation (3).

$$p_i(x) = \frac{1}{(2\pi)^{\frac{d}{2}} \sigma^d} \frac{1}{N_i} \sum_{j=1}^{Ni} \exp[\frac{(x - x_{ij})^T (x - x_{ij})}{2\sigma^2}] \qquad (3)$$

$$\hat{C}(x) = \text{argmax}\{p_i(x)\}, i = 1, 2\ldots, m \tag{4}$$

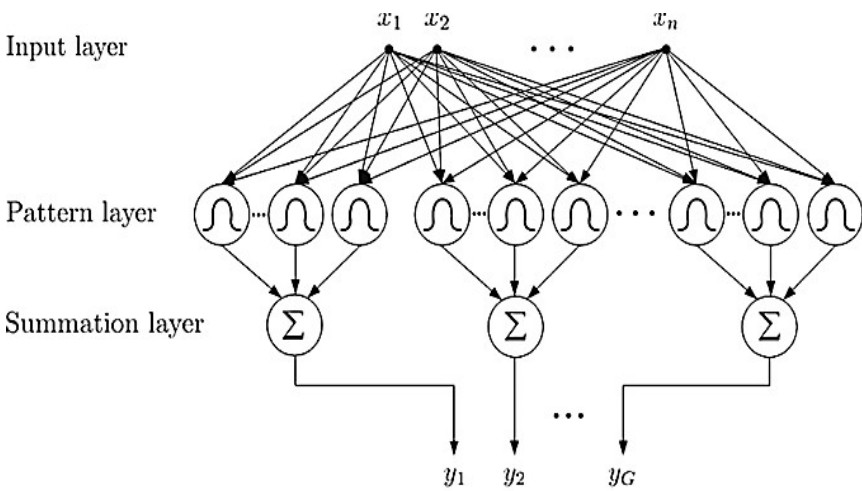

**Figure 2.** Architecture of PNN.

Based on the Bayes criterion, the final output is computed using Equation (4). Where $C(x)$ represents the class estimation with respect to the pattern $x$. $m$ denotes the total samples in a class.

*3.5. Smoothing Parameter Selection*

The performance of the proposed approach is based on the PNN for SQL injection attack detection and relies on the smoothing parameters, which show the distribution spread and define how many data are utilized to fit each local polynomial. Regardless of the complex nature of the PNN, it only has one training parameter in the form of a smoothing parameter. It is crucial that a smoothing parament must have the best value for performance enchainment. Obtaining the best value for the smoothing parameters can be achieved through either a numerical or heuristic-based natural technique. Conventionally, the best value of soothing paraments was achieved through the hit and error method. The downside of using the hit and error method is that it is a very time-consuming procedure that will affect the neural network's performance. We employ the BAT algorithm in this approach, a nature-inspired optimization algorithm based on the bat developed by [32]. Compared to other optimization methods, this one can find nearly optimal solutions to hard optimization problems in a reasonable amount of time. This is because the BAT algorithm uses a well-balanced approach to exploration and exploitation to search the solution space and avoid local optima. The BAT algorithm can be used to solve many optimization problems because it is easy to set up, flexible, and doesn't need many tuning parameters. The BAT algorithm also has the ability to work in noisy and dynamic situations, which are typical of real-world issues. Overall, the BAT algorithm is a powerful method of optimization that has many benefits, such as being effective, adaptable, and resilient.

A bat typically expands its sensing capabilities to detect prey efficiently, and this behavior guides it towards the articulation of an optimization algorithm and it can be employed for solving a real-time problem. Suppose a bat $B$ at a certain position $p_i$ flies with a velocity $v_i$ to hunt prey, with a changing wavelength $\lambda$ and a loudness $A_0$ even in complete darkness [33]. Algorithm 1 defines the working of a standard bat algorithm obtained from [34].

---

**Algorithm 1:** Standard Bat Algorithm

---

Begin

    Initialize position, velocity and other parameters for each bat

    While (Stop criteria is met?)

        Randomly generates the frequency

        Update the velocity

        Update the position

   If rand<$r_i^t$

    Update the position

  End

  Calculate the fitness;

   If (rand<$A_i^t$)&& ($f\left(x_i^t\right)< f(x^*)$)

    Replace the position with the new one

     Update $r_i^t$ and $A_i^t$

   End

    Select the current global best position

  End

    Output the best position

End

---

*3.6. Model Evaluation*

In this study, the evaluation of the model is performed with the help of a confusion matrix. The confusion matrix provides a powerful statistical tool for measuring the model's performance by evaluating binary classifiers. It enables one to determine how well a classifier conducted testing on a held-out set is. If the classifier is performing poorly or the model is not able to be understood, then it is likely that the model will perform poorly on a test set. The best way to catch and improve upon these types of errors is to perform an analysis. This involves taking the test set predictions and comparing them with actual values. After processing the test set, we can see that our classifier has generated some false positives and negatives. Given this information, we might decide to retrain our model or make a few tweaks to improve the classifier's performance. Confusion matrices for a classifier are typically displayed as cells in a 2 × 2 grid. The first row describes the "true state" of each example in the dataset, and the second row describes the "predicted state" based on the classification rule as given in Figure 3.

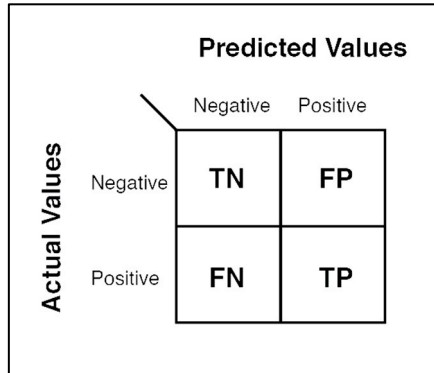

**Figure 3.** Confusion Matrix.

Based on the confusion matrix, we used four evaluation methods to evaluate the proposed model, including the accuracy, precision, recall, and F-Measure, as given in Equations (5)–(8).

$$Accuracy = \frac{TP + TN}{TP + TN + FP + FN} \tag{5}$$

$$\text{Precsion} = \frac{\text{TP}}{\text{TP} + \text{FP}} \tag{6}$$

$$\text{Recall} = \frac{\text{TP}}{\text{TP} + \text{FN}} \tag{7}$$

$$\text{F} - \text{Measure} = 2 \times \frac{\text{Precesion} * \text{Recall}}{\text{Precesion} + \text{Recall}} \tag{8}$$

## 4. Experimental Results and Implementation Details

The experiments were conducted using a PC with an Intel(R) Core i7-7500 CPU @ 2.70 GHZ, 2 Core(s), 4 Logical Processor(s) with 4 GB of primary memory, and the whole environment was simulated using the python programming language and the PyTorch deep learning framework [35]. The dataset used in the proposed approach comprised malicious and benign network traffic generated from a virtual traffic server designed to mimic a real environment. The dataset included 6000 malicious and 3500 benign SQL queries, captured inbound to a web application that included features such as TCP and HTTP information about packets. The web traffic also carried malicious and benign SQL injection attack queries. The SQL features were generated using a tokenizing method, such as token count, token type, and token value. The features extracted for this study were selected by employing the Chi-Square testing method to obtain such features that would contribute towards high accuracy; these were retained and are given in Table 1.

In this study, a 10-fold cross-validation scheme was employed to distribute the data equally for training and testing. A k-fold cross-validation scheme divides the original dataset into a training set to train the model, and a testing set to test the predictive model. The range of k varies from 1 to $k - 1$. The 10-fold cross-validation divides the entire dataset into 10 distinct data subsets. During the experiment for each data subset, the remainder of the 9 subsets are used to train the classifier, and the final subset is taken as a final test set. The results obtained from the experiments using the proposed approach, and the comparison with other machine learning algorithms, are shown in Table 2.

Since we used the BAT algorithm as the smoothing parameter estimation, which helped us to achieve the best value and great prediction accuracy, the parameter settings used in the approach were given as:

1.  The bat size was limited to 30, and the number of iterations was set to a maximum of 100, as represented in Equation (9).

$$T_{max} = 100 \tag{9}$$

2.  For each bat, the position was calculated randomly. The position of the bat gave the smoothing parameter *h*.
3.  The bat's primary position was randomly generated with a uniform distribution of [0, 10].
4.  The fitness function *ff*, as given in Equation (10), was used; this takes the induvial solution of a problem as the input and generates the output to define "how fit and good" the solution is with respect to the actual problem.

$$ff = min\frac{1}{n}\sum\nolimits_{i=1}^{n}\left(y_i - \hat{y}_i\right)^2 \tag{10}$$

5.  The position of the bat was updated to:

$$B_{new} = B_{old} + A_0 \tag{11}$$

6.  Steps 4 and 5 were repeated until $T_{max}$ was achieved.

The experiments were conducted in three scenarios based on the data partition, as given in Tables 2–4. All the classifiers' parameters were kept the same, and the feature size was also kept the same. The experimental results show that the PNN in all three scenarios performed better. While using the 10-fold cross-validation, an accuracy of 99.19% was achieved compared to all classifiers in all three scenarios.

Scenario 1: Using 100% Training Data.

**Table 2.** Results obtained using 100% Training.

| Classifier | Accuracy | Precision | Recall | F-Measure |
|---|---|---|---|---|
| PNN | 98.69% | 0.975% | 0.961% | 0.964% |
| SVM | 95.62% | 0.95% | 0.884% | 0.878% |
| Decision Tree | 96.31% | 0.917% | 0.926% | 0.916% |
| ANN | 97.25% | 0.984% | 0.876% | 0.932% |

Scenario 2: Partitioning the data into 80:20 Training and Testing.

**Table 3.** Results obtained using 80:20 partition.

| Classifier | Accuracy | Precision | Recall | F-Measure |
|---|---|---|---|---|
| PNN | 98.11% | 0.978% | 0.972% | 0.917% |
| SVM | 96.65% | 0.942% | 0.975% | 0.978% |
| Decision Tree | 97.42% | 0.96% | 0.889% | 0.881% |
| ANN | 98.18% | 0.984% | 0.876% | 0.932% |

Scenario 3: Using 10-Fold Cross Validation.

**Table 4.** Results obtained using 10-Fold Cross Validation.

| Classifier | Accuracy | Precision | Recall | F-Measure |
|---|---|---|---|---|
| PNN | 99.19% | 0.995% | 0.981% | 0.928% |
| SVM | 96.32% | 0.96% | 0.889% | 0.881% |
| Decision Tree | 97.33% | 0.972% | 0.968% | 0.938% |
| ANN | 98.54% | 0.98% | 0.963% | 0.924% |

## 5. Conclusions

SQL injection attacks continue to be one of the topmost security challenges affecting financial, health, and other essential data. The challenge of detecting SQL injection attacks has increased its importance as our dependence on the internet is growing exponentially. The main objective of this study was to investigate the effectiveness of the proposed PNN for SQL injection detection. In this study, it has been established that using the BAT algorithm for optimization helped achieve high accuracy. The selection of the best smoothing parameter enabled the PNN to achieve an accuracy of 99.19% with a precision of 0.995%, a recall of 0.981%, and an F-Measure of 0.928%. Compared to the studies [21,22,24], the proposed approach performed better. The advantage of this study is that high accuracy, fast performance, and low false positive rates have been achieved. However, the main challenge is the high complexity and the sensitivity to noise on irrelevant features.

**Author Contributions:** Conceptualization, F.K.A. and N.A.K.; Methodology, F.K.A. and N.A.K.; Validation, F.K.A.; Formal analysis, F.K.A. and N.A.K.; Investigation, N.A.K.; Writing—Original draft, F.K.A. and N.A.K. All authors have read and agreed to the published version of the manuscript.

**Funding:** This work was supported by the Deanship of Scientific Research, Vice Presidency for Graduate Studies and Scientific Research, King Faisal University, Saudi Arabia, [Grant No. 2896].

**Institutional Review Board Statement:** Not applicable.

**Informed Consent Statement:** Not applicable.

**Data Availability Statement:** https://pypi.org/project/libinjection-python (accessed on 1 March 2023).

**Conflicts of Interest:** The authors declare that they have no conflict of interest to report regarding the present study.

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
