# Peer review of "Enhancing the Performance of SQL Injection Attack Detection through Probabilistic Neural Networks"

_applsci, doi:10.3390/app13074365_

Round 1

Reviewer 1 Report

After reviewing the manuscript, several recommendations have been made to improve the quality of the study. These include addressing the points mentioned below. Despite this, it is noted that the topic and title of the manuscript are well-explained. Therefore, if these recommendations are implemented, the quality of the manuscript will be significantly improved.

1.      The main issue and problem for which a solution is proposed needs to be better explained in the introduction.

2.      The authors should provide more detail and recent issues on SQL Injection attacks.

3.      More detail on existing research that uses machine learning techniques to detect previously unknown and novel attack types should be included.

4.      Provide more detail problems on the conventional defence approaches.

5.      Provide more details about the most dangerous vulnerabilities exploited to leak sensitive information, unauthorized access, and cause financial loss to individuals and organizations.

6.      Taking advantage of machine learning, a probabilistic neural network (PNN) is proposed to improve detection accuracy of SQL injection attacks.

7.      The authors should provide more details about advantages of using BAT algorithm, a metaheuristic algorithm for optimization.

8.      Carefully check equation numbering.

9.      There are some typo and grammatical errors which needs to be corrected such as “students”.

10.  What is scheme 25?

11.  Some refences are incomplete such as [2], [3].

12.  Properly follow Journal format guidelines.

Reviewer 2 Report

The article is devoted to building an efficient and accurate model for SQL injection attack detection. Authors proposed a probabilistic neural network (PNN) to detect SQL injection attacks. To achieve the best value in selecting a smoothing parament, they employed the BAT metaheuristic algorithm. In the current version, it is difficult to understand the authors' contribution to the problem field. There are also several issues that need to be clarified:

1. There are many metaheuristic algorithms, each aimed at solving certain problems, that may be good in their area but show average results even in related areas. Justify the choice of this particular metaheuristic algorithm? Was it modified to solve your problem? - Judging by the given algorithm, it was used in the standard version; in which case were the parameters tuned? 

2. In your work, you apply a metaheuristic algorithm as an optimization algorithm. Such algorithms are known to be able to work in combination with feature selection and show good results. Why don't you add some different (classical) feature selection methods in addition to the standard one used in the algorithm? This is  especially true since you have a feature selection module and use metaheuristics for optimization, which could optimize the feature space.

3. There are three root causes of SQL injection vulnerabilities: the combining of data and code in a dynamic SQL statement, error revealation, and insufficient input validation. How does your model deal with causes unrelated to the construction of the query itself that cannot be traced syntactically, namely flaws in the system itself - broken authentication and broken access control? Statistically, these causes are a big part of the injection phenomenon. 

4. What steps would you recommend using in conjunction with your system to detect injections? 

5. Is your model able to handle different SQL dialects? How is this achieved?

6. Unify the metrics. Accuracy is used as a percentage, and the others are not. Reduce them to the same form. 

7. To assess the authors' contribution to the subject area, add a discussion section in which you describe the advantages and disadvantages of your proposed approach compared to other works reviewed in Related Works.

Round 2

Reviewer 2 Report

The authors have solved almost all the doubts that I had with this work and have included those explanations in detail in the article. However, I still do not understand the reasoning of the metaheuristic choice, and in the paper it does not declare. In addition, explain the choice of standard metaheuristic parameters wihout any attempt to carefully set it for your problem?
